

**Characteristics of vertical velocities estimated from drop size and fall**
**velocity spectra of a Parsivel disdrometer**
Dong-Kyun Kim and Chang-Keun Song
*School of Urban and Environmental Engineering,*
*Ulsan National Institute of Science and Technology, Ulsan, Korea*

Corresponding Author: Prof. Chang-Keun Song, School of Urban and Environmental Engineering, Ulsan National
Institute of Science and Technology, Ulsan, Korea, Email:cksong@unist.ac.kr





Abstract

Vertical air velocities were estimated from drop size and fall velocity spectra observed by
Parsivel disdrometers during intensive field observations from 13 June to 3 August 2016
around Mt. Jiri (1915 m above sea level) in the southern Korean Peninsula. Rainfall and wind
velocity data measured by Parsivel disdrometers and ultrasonic anemometers, respectively,
were analyzed for an orographic rainfall event associated with a stationary front over Mt. Jiri
on 1 July 2016. In this study, a new technique was developed to estimate vertical air
velocities from drop size and fall velocity spectra measured by the Parsivel disdrometers and
investigate characteristics of up-/downdrafts and related microphysics in the windward and
leeward side of the mountain.
To validate results from this technique, vertical air velocities between the Parsivel and
anemometer were compared and were in quite good agreement each other. It was shown that
upward motions were relatively more dominant in the windward side and even during periods
of heavy rainfall. On the contrast, downward motions were more dominant in the leeward
side even when heavy rain occurred. Occurrences of upward and downward motions were
digitized as percentage values as they are divided by the total rainfall period. In the windward
(leeward) side, the percentages of upward (downward) motion were larger than those of
downward (upward) motion. Rainfall intensity in the leeward side was relatively stronger
than in the windward side, suggesting that the increase in rainfall in the leeward side was
more affected by the downward motions.



Key words: vertical air velocity, drop size spectra, microphysics, Parsivel disdrometer



## 1. Introduction

Drop size distribution (DSD) and related rain parameters from surface disdrometer
measurements or indirectly retrieved from remote sensing measurements such as radars, wind
profilers, or satellites provides key information for a better understanding of microphysical
processes that account for drop growth or decay within precipitating systems. However, DSD
uncertainties always exist as its retrieval is vulnerable to various factors such as measurement
errors, sampling difference in volume and height, strong winds, up-/downdrafts, turbulence,
and so on as have been reported in many previous studies (Jameson and Kostinski 1998; Cao
et al., 2008; Tokay et al., 2009; Thurai et al., 2012). Thus a validation of such retrieved DSDs
by comparing with those from surface disdrometers is not straightforward (Williams et al.
2000) due to their different environment although minimizing a sampling difference as much
as possible is needed. Even if DSDs are accurately obtained, their characteristics, particularly
between convective and stratiform rain, can vary largely from small areas in short-time scale
to climatic regimes in long-term.
Ground-based classifications of convective, mixed, or stratiform rain type have been
performed in various ways such as characteristics in integral DSD parameters (i.e., rain rate,
mean drop diameter, etc), bright band signature, vertical gradients in Doppler velocity and
reflectivity, vertical draft magnitude, and so on (Atlas et al., 2000; Cifelli et al., 2000; Tokay
et al., 1996, 1999; Thurai et al., 2016; Williams et al., 1995). Tokay et al. (1999) classified
rainfall types from collocated disdrometer and 915 MHz profiler observations in tropical rain
events and indicated that compared to profiler classifications that utilize vertical gradients in
Doppler velocity, a disdrometer is relatively more feasible to misclassify stratiform rain as
convective or vice versa due to time-height ambiguity mostly associated with advection of
drops while falling to the ground.
In measuring and validating surface DSDs, there is no such handy, transportable, and low-



cost instrument like disdrometer that has long been used as a ground truth although it has
inherent problems mentioned above as exposed to all different environments. Parsivel
disdrometer (hereafter Parsivel) is one of disdrometers widely used for DSD studies over the
world. As deduced from its name, par-si-vel (particle size and velocity), this disdrometer
measures fall velocities, sizes, and number counts of liquid and ice particles falling into 32
(size) x 32 (fall velocity) bins. Parsivel has been used at observatories or in numerous field
experiments to examine and validate microphysical properties by comparing DSDs and
integral DSD parameters with those from other type disdrometer, 2-Dimentional Video
Disdrometer (2DVD) and radar and profiler observations for various events of precipitation
(Kim et al., 2010; Thurai et al., 2016).

A Parsivel-measured fall velocity of a raindrop is the sum of a raindrop terminal fall

speed (in stagnant air) and vertical air motion. Thus when there are updrafts or downdrafts,
the Parsivel-measured fall velocity is deviated from the terminal fall speed even if drop sizes
are identical. On top of this, strong horizontal winds, vertical shear, or turbulence can
disperse the distribution of drop size and fall velocity, leading to a change (or bias) in the
Parsivel-measured fall velocity averaged over the distribution. Consequently, all these factors
would affect DSD integral parameters such as rain rate although single or mixed effects of the
factors on these have not been fully understood (Niu et al., 2010). Ulbrich (1992) examined
errors in rain rate that result from inaccuracies in fall speeds of raindrops (i.e., inaccurate
estimation of vertical air motion) and indicated that updraft will result in larger rain rate at a
given reflectivity than when there are no vertical winds. Niu et al. (2010) investigated
differences in distributions of drop sizes and fall velocities between convective and stratiform
rain and ascribed different deviations in Parsivel-measured fall velocities between small and
large drops to vertical air motion and turbulence. Parsivel is prone to measurement errors
particularly when there are strong winds and turbulence, leading to discrepancies in


comparison with other measurements in the same locations. Tokay et al. (2009, 2014)
indicated that the old version of Parsivel tends to underestimate the number of small drops
and overestimate drop size larger than 2.0 mm in heavy rain as well as in windy conditions.
When they compared each old and new version of Parsivel with Joss-Waldvogel disdrometer
and rain gauge measurements, the new version of Parsivel (referred to as Parsivel[2] in their
paper) appeared to have a noticeable improvement over the old one for measuring drop size
and rainfall rate.
To our knowledge, no studies of vertical air velocities retrieved from Parsivel-measured
drop size-fall velocity spectra have been documented or reported yet. In this study we utilize
Parsivel and anemometer data collected during intensive field observations that targeted to
investigate orographic rainfall mechanisms around mountain areas in the southern region of
Korea. A simple technique to retrieve vertical air velocities from Parsivel measurements is
developed and first applied to an orographic heavy rain event. This paper is organized as
follows. In Section 2, the retrieval technique and instruments used in this study are introduced.
A case description about the rain event is followed in Section 3. Results about characteristics
of up-/downward motions and related microphysics in the windward and leeward side are
presented in Section 4. A summary and conclusions follow in Section 5.

## 2. Instrumentation and method

Two main instruments used in this study are Parsivel disdrometer and ultrasonic
anemometer collocated at three different stations around Mt. Jiri (see Figure 1). Their data
were collected during the intensive observation period from 13 June to 3 August 2016 to
cover a summer rainy season which is called "Changma" in Korea were analyzed. Parsivel
disdrometer (Parsivel), manufactured by OTT (Germany), uses laser-optical properties to
measure both sizes and fall velocities of precipitation particles and derives quantities of radar



reflectivity, precipitation intensity, etc from measured drop spectra. Time resolution is 1 min.
For more details about Parsivel, please see the Löffler-Mang and Joss (2000)'s paper. The
ultrasonic anemometer (hereafter UVW) measures east-west ($u$), north-south ($v$), and vertical
($w$) components of winds by using the speed of sound moving along winds between the three
nonorthogonal sonic axes and generates wind speed and direction at 1-min interval. The $w$
component observed by UVW is referred to as $w_{UVW}$.

In this study, a simple, new scheme to derive vertical air velocity ($w$) from Parsivel

measurements is developed by using a relationship of Atlas et al. (1973) between terminal
fall velocities and drop diameters in still air as shown by
$$V_f = 9.65 - 10.43 * \exp(-0.6D), \qquad\qquad (1)$$
where $D$ is drop diameter in mm and $V_f$ is terminal fall velocity (m s$^{-1}$) and also the vertical
relation of air as shown below
$$w = V_p - V_f (\rho_0/\rho)^{0.4} \qquad\qquad (2)$$
where $V_p$ is Parsivel-measured fall velocity averaged over 32 diameter classes in a size and
velocity spectrum and $(\rho_0/\rho)^{0.4}$, where $\rho$ is air density (kg m$^{-3}$) parameterized by $\rho(z) = \rho_0 \exp($-
$z/9.58)$ and $z$ is altitude in kilometers, is the term that corrects $V_f$ in relation to atmospheric
density (Beard 1985) since $V_f$ increases as atmospheric density decreases exponentially with
height. In all the terms, negative means downward. The atmospheric density correction was
applied to calculate $w$ although the altitudes of D1, D2, and D4 are relatively low at 105, 280
and 313 m AGL, respectively. Thus a mean $w$ value at 1-min interval is finally estimated by
subtracting $V_p$ from $V_f$ calculated from Eq (1). The final $w$ estimate is hereafter called $w_{par}$.
For more details, please see the flowchart in Figure 2 that shows how $w$ is estimated from a
1-min drop size ($D$) and fall velocity ($V_p$) spectrum of Parsivel. Figure 3 illustrates three
conditions of determining zero $w$, upward $w$, or downward $w$ value for given size and fall



velocity spectra. For the case 1, $w$ would be zero since the $D$-$V_p$ distribution closely follows
the $V_f$ line. Upward $w$ value is determined for the case 2 that $V_p$ is smaller than $V_f$ (i.e., the
distribution is towards below the $V_f$ line). For the case 3, downward $w$ value is determined
since $V_p$ is larger than $V_f$. For $w_{par}$ validation, $w_{par}$ is compared with $w_{UVW}$ and its result is
described in Section 4.

**3. Case description**
During a summer rainy season usually from late June to mid July in Korea, severe
weather phenomena accompanied by heavy rainfall often occur in the southern region of the
Korean Peninsula mostly covered by complex high mountains. In association with terrain-
induced up-/downdrafts, mountainous areas can play an important role in controlling
formation, amount, and distribution of rainfall. As precipitation systems move over these
areas, they tend to develop rapidly and produce localized heavy rainfall. Observational
analysis from radar and surface measurements in these areas is necessary to understand
terrain effects on rainfall development and microphysics. Thus we performed intensive field
observations around Mt. Jiri (1915 m ASL) during the 2016 summertime in the southern
Korean Peninsula.
During the observation period of 13 June~3 August 2016, several rain events were
observed. On 1 July 2016, a rainfall system associated with a Changma front has developed
over the West Sea and moved towards Mt. Jiri. As it passes over the mountain from the east,
heavy rainfall was produced and observed by Parsivel disdrometers and UVWs from 1200 to
2200 UTC. Also a dual-Doppler radar analysis (Liou et al., 2012) was also conducted to
obtain 3-D wind components from radial velocity data of two Doppler radars as well as
vertical structure of radar reflectivity in this mountain area. Figure 4 shows a daily
accumulated rainfall distribution and topography of Mt. Jiri. Large rainfall up to 90 mm was



seen around the top and south of Mt. Jiri. This is related to more moist upwind flows in the
windward side closer to the ocean.

**4.    Results**
*4.1. w comparison in time series*

For the $w_{par}$ validation, the observed $w_{UVW}$ is compared at time series. Time series of radar

reflectivity ($Z$), rain rate ($R$), mass-weighted mean diameter ($D_m$) measured from Parsivel are
also examined together. As shown in Fig. 4, three stations of D1, D2, and D4 where both the
Parsivel and UVW data are available were selected out of total nine stations. D1 and D2 are
windward and D4 is leeward of Mt. Jiri. Figure 5 shows the time series of $Z$, $R$, and $D_m$ (left)
and $w$ (right) between the Parsivel and UVW observed at D1, D2, and D4. At D1 and D2,
high values of $Z > 40$ dBZ and $R > 20$ mm h$^{-1}$ are observed during the 1230-1330 UTC period
and at around 1730 UTC in Figs. 5a and b. Correspondingly, large $D_m$ values reaching 2 mm
were analyzed in these periods. In Fig. 5c, high $Z$ and $R$ were also observed in the leeward
side but showing a little time lag compared to those in Figs. 5a and b.

It is shown in Figs. 5d, e, and f that $w_{par}$ matches quite well with $w_{UVW}$. On the windward

side (D1, D2), they both show mostly upward motions and importantly, larger upward
motions during periods of heavy rainfall (i.e., 1230-1330 UTC and around 1730 UTC). In
contrast, downward motions are mostly found on the leeward side. It is noted that there exists
relatively large difference between $w_{par}$ and $w_{UVW}$ with the opposite signs during these high $R$
periods in Fig. 5f. The upward biases of $w_{par}$ in these periods are related to the fact that for a
given $V_f$, a mean $V_p$ became smaller in Eq (2) due to an increase of a larger number of drops
ranged at 1~3 mm or a scatter of these drops below the $V_f$ line in the $D$-$V_p$ distribution, to be
more like the case 2 in Fig. 3. A physical reason for this is not clear yet but it is probably
resulted from strong winds and turbulence during the high $R$ periods. In other periods, they



show quite good agreement. Also, the maximum values of $w_{par}$ and $w_{UVW}$ hardly exceed $\pm 0.5$
m s$^{-1}$, almost the one-fifth of horizontal wind magnitudes (not shown), suggesting that winds
are almost horizontal during the analysis period and they just pointed up or downward,
depending on $w$ signs and magnitudes. At D1 and D2, the relatively larger $w_{par}$ and $w_{UVW}$
were found during heavy rain with $R > 20$ mm h$^{-1}$ around 1300 and 1740 UTC (Figs. 5d and
e), indicating that updrafts contributed more on the substantial $R$ increase on the windward
side. As downward motions were found on the leeward side even during the periods of heavy
rain ($R > 20$ mm h$^{-1}$) as shown in Fig. 5f, an $R$ increase on the leeward side is more associated
with the downward $w$ component of winds.

Figure 6 shows characteristics of $Z$-$R$ relations at D1, D2, and D4. The upward $w_{par}$

values are colored in red and the downward $w_{par}$ in blue. Also we converted them into
percentages by dividing by a total of counts in each class with $R > 0.5$ mm h$^{-1}$ only. At D1
and D2, they were similar as the percentage for the upward $w_{par}$ class is 61% and 71% and
the percentages for downward $w_{par}$ class is 39% and 29%, respectively. In contrast, the
upward $w_{par}$ percentage at D4 is 31%, much smaller than that at D1 and D2 as found in Fig. 5,
and the downward $w_{par}$ percentage is 69%. It is also important to see subtle changes in a
coefficient $\alpha$ and exponent $\beta$ in a $Z$-$R$ power law form of $Z=\alpha R^{\beta}$. There was a decrease in $\alpha$
from D1 and D2 (250, 252) on the windward side to D4 (226) on the leeward side and a slight
increase in $\beta$ from D1 and D2 (1.31, 1.39) to D4 (1.43). This makes the $Z$-$R$ line less steep
and thus for a given $Z$, $R$ is larger at D4 than D1 or D2, indicating that rainfall are relatively
more strong at D4.

*4.2. Histogram analyses*

4.2.1 $w$ histograms with regard to $R$

The $w_{par}$ and $w_{UVW}$ time series discussed in Section 4.1 are examined in their histograms



with regard to $R$. In this study, stratiform and convective rain was simply classified by a
threshold of $R < 10$ mm h$^{-1}$ and $R > 10$ mm h$^{-1}$, respectively, which has been often used in
previous DSD studies (Leary and Houze 1979). In Figs. 7a, b, c, the $w_{par}$ histograms showed
overall agreement with the $w_{UVW}$ at all stations and relatively, the better agreement was found
in the stratiform class ($R < 10$ mm h$^{-1}$) than the convective class. D1 and D2 showed the quite
similar histograms of $w_{par}$, compared to D4. At D1 and D2, all convective rain has occurred
almost in association with upward motions, while for stratiform rain, it occurred with both
upward and downward motions (Figs, 7a and b). Importantly, almost the half or a little larger
portion of stratiform rain has occurred in association with upward motions. In contrast, at D4,
most of stratiform rain was associated with downward motions and convective rain was
associated with both upward and downward motions (Fig. 7c). In other words, the latter
indicates that heavy rainfall on the leeward side was relatively more associated with
downward motions than on the windward side (D1, D2) as noted in the previous section.
Figures 7d,e,f show the areas occupied by the upward and downward $w$ values in percentage
at each station, same as those shown in the $Z$-$R$ scatterplots in Fig. 6. Dominant $w$ class is
easily seen by these percentage values at each station. In the downward $w$ group, the largest
percentage (69%) is found at D4 (Fig. 7f).

4.2.2 Characteristics of $Z$ histograms with regard to $w$ and $R$
The $w_{par}$ properties discussed in Section 4.1 are examined by $Z$ histograms with regard to
$w$ and $R$. In Fig. 8a, a much larger percentage (61%) in the upward $w$ group is found at D1
showing a relatively wider $Z$ distribution, compared to that at D4 in Fig. 8d. In Fig. 8b, the $R$
percentage classified as convective was 9%, much smaller than 61% in the upward $w$ group
in Fig. 8a, suggesting that 52% of the upward $w$ group was associated with stratiform rain. In
order to study such relationships between $w$ and $R$, histograms were split by four conditions





in the upper-right corner shown in Figs. 8c and f. That is, each group of $R > 10$ mm h$^{-1}$ and $R$
$< 10$ mm hr$^{-1}$, which is regarded as convective and stratiform rain, respectively, is separated
by upward and downward $w$. Therefore, for instance, 91% of the group $R > 10$ mm h$^{-1}$ in Fig.
8c is equal to the sum of 52% of the upward $w$ and 39% of the downward $w$ group. Likewise,
the upward and downward $w$ group is also split by the two $R$ conditions. Unlike D4, there
was no thick blue line at D1 in Fig. 8c because there were no data fell into this category of
the downward $w$ and $R > 10$ mm h$^{-1}$ as shown in Fig. 7a.

In Fig. 8c, convective rain ($R > 10$ mm h$^{-1}$) with the largest mean $Z$ has occurred solely in

association with upward $w$ motions (thick red line). Among the four categories, the majority
percentage of 52% was found in the category of the upward $w$ and $R < 10$ mm h$^{-1}$ at D1 but
65% was found in the category of the downward $w$ and $R < 10$ mm h$^{-1}$ at D4. The widest $Z$
distribution were shown in these categories. In Fig. 8d, a much larger percentage is found in
the downward $w$ group as noted previously. In Fig. 8e, a larger percentage of 18% is found in
the group $R > 10$ mm h$^{-1}$, compared to the counterpart (9%) at D1, indicating that on average
sense, rain intensity was stronger at D4 (leeward). It is noted that at D4, convective rain has
occurred in association with both updrafts (14%) and downdrafts (4%). The little portion of
convective rain with relatively smaller $Z$ happened in association with downdrafts. It is thus
suggested that downdrafts can play a significant role in increasing $R$, even larger than 10 mm
h$^{-1}$ although the strongest $R$ was related to updrafts rather than downdrafts. Most of stratiform
rain was associated with downdrafts (65%).

4.2.3. Histogram characteristics of DSD parameters with regard to $w_{par}$

In Fig. 9, characteristics of DSD parameters retrieved from the Parsivel measurements are

analyzed with regard to $w_{par}$. Compared to classic DSD retrieval studies without considering
$w$ properties, in this study, two histograms split by the upward and downward $w$ were



obtained *per* each parameter, which is a first time ever. In Fig. 9b, The $Z$ histograms at D4
show higher $Z$ distributions with mean values of 34.8 and 25.6 dBZ in the upward and
downward $w$ category, respectively, are shown, compared to those (25.2 and 18.2 dBZ) at D1
in Fig. 9a. At both D1 and D4, the mean $Z$, $R$, and $D_m$ values in the upward $w$ category were
higher than those in the downward $w$ category. Between D1 and D4, the mean $Z$, $R$, and $D_m$
over the entire data were higher at D4, indicating that rain intensity was somewhat stronger
than D1 although the maximum $Z$ (~50 dBZ) and $R$ (below ~60 mm hr$^{-1}$) were quite similar
each other (see the time series of $Z$ and $R$ in Fig. 6). For $R$, the mean $R$ value of 15.1 mm hr$^{-1}$
was higher at in the upward $w$ group at D4 than 6.22 mm hr$^{-1}$ in the counterpart at D1 (Figs.
9c and g). The mean $D_m$ was largest at 1.37 mm in the upward $w$ category of D4 in Fig. 9f
and smallest at 0.86 mm in the downward $w$ category of D1 in Fig. 9b. Thus, the mean $D_m$
(1.03 mm) in the downward $w$ category of D4 was greater than that (0.86 mm) in the same
category of D1. This indicates that there were a comparatively larger number of large drops at
D4 in association with downward motions which were dominant even during the strong $R$
period. Thus, it is stressed that relative to the windward side, downward motions have more
influenced the growth in drop size and increase in $R$ intensity in the leeward side.

**5.   Summary and conclusions**

Intensive field observations around Mt. Jiri for the orographic rainfall events associated

with a Changma front in the southern regions of the Korean Peninsula were conducted during
summertime in 2016. In order to examine up-/downward $w$ properties in the windward and
leeward side of Mt. Jiri, a simple technique was newly developed to retrieve vertical
velocities ($w$) from drop size and fall velocity spectra of the Parsivel disdrometers at different
stations. Their comparison with the $w$-components observed by UVW showed quite good
agreement each other, producing the similar histograms between the two instruments. On the





windward side (D1 and D2), upward motions were more frequently observed and particularly
larger upward motions were found during the periods of strong $R > 10$ mm h$^{-1}$. For the
leeward side (D4), downward motions were more dominant even during the same strong $R$
periods as in the windward side. The downward $w$ percentage was more than a double of the
upward $w$ percentage. Importantly, this suggests that strong rain of $R > 10$ mm hr$^{-1}$ in the
leeward side was more associated by negative $w$-components of winds. Most of stratiform
rain also occurred with downward motions. Thus, compared to the windward side where
upward motions dominated, downward motions have contributed more on drop growth as
well as $R$ increase in the leeward side.

Therefore, the newly developed technique to estimate $w$ values from Parsivel drop size

and fall velocity spectra is found physically meaningful and can be applicable to $w$ studies to
compare with other $w$ retrievals and investigate their effects on rainfall development in
relation to up-/downdrafts aloft especially in orographic regions. The estimated $w$ values
showed notably different characteristics in magnitude and signs between the windward side
and leeward side during rainfall with dependence on how a size and fall velocity spectrum
distributes. Thus, the observed $D$-$V_p$ distributions appeared to be strongly connected to $w$
properties at surface. In this study, the estimated $w$ values are small in magnitude mostly less
than $\pm0.5$ m s$^{-1}$, almost the one-fifth of horizontal wind magnitudes, indicating that
prevailing winds were almost horizontal in this case study.

There was a relatively large difference between $w_{par}$ and $w_{UVW}$ with the opposite signs

during these high $R$ periods (Fig. 5f). This is probably associated with strong winds and
turbulence that may scatter the $D$-$V_p$ distribution of drops below the $V_f$ line and further bias $w$
magnitudes. Hence the $w$ retrieval from the disdrometer-based technique is not totally free
from environmental conditions. Since the effects of winds and turbulence were not analyzed
in this study, we will soon investigate their effects on $D$-$V_p$ distributions as well as resultant $w$



biases in a quantitative way as a subsequent work.


















**Acknowledgements**
This research was supported by the National Strategic Project-Fine Particle of the National
Research Foundation of Korea(NRF) funded by the Ministry of Science and ICT(MSIT), the
Ministry of Environment(ME), and the Ministry of Health and Welfare(MOHW).(NRF-
2017M3D8A1092021)





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




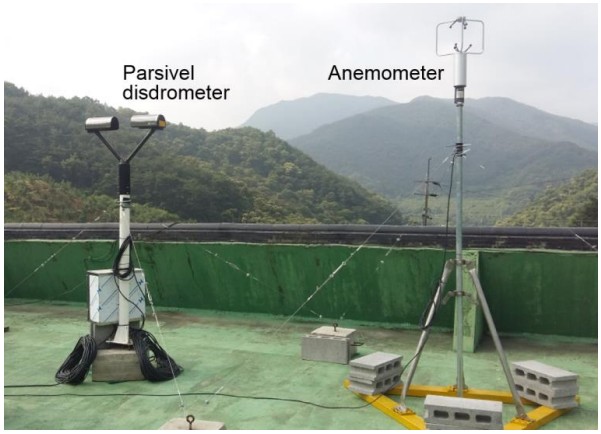

Figure 1. Picture of a Parsivel disdrometer and 3-D anemometer that were installed at a station around
Mt. Jiri during the intensive observation period.






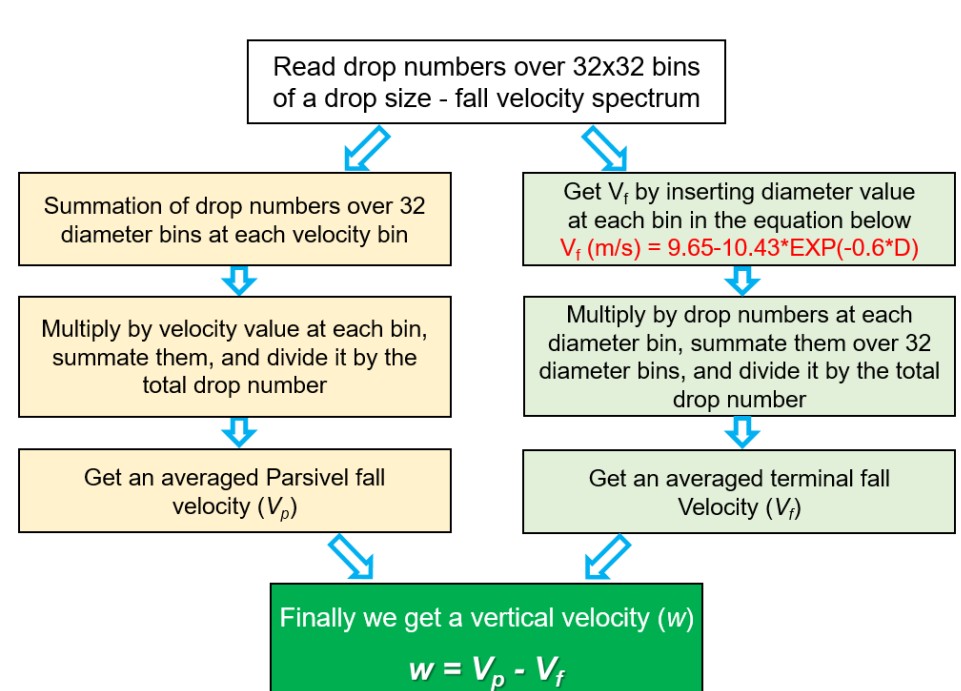

Figure 2. Flowchart for estimating *w* from a diameter-fall velocity spectrum of Parsivel (1-min
interval). See text for more details.






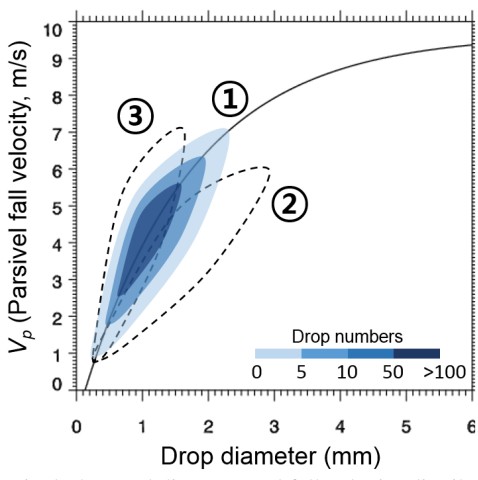

Figure 3. Schematic of Parsivel-observed diameter and fall velocity distributions for the three cases of
determining zero $w$, upward $w$, and downward $w$. Contours show drop number concentrations. See
text for more information.




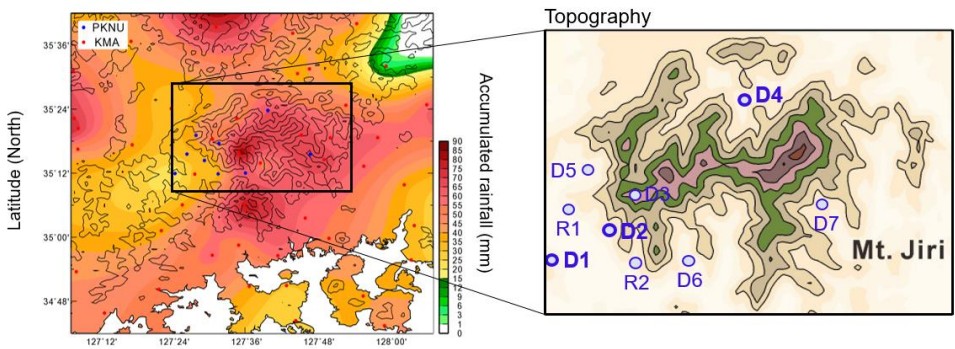

Figure 4. (a) Distribution of an acumulated rainfall (mm) on 1 July and (b) the topography of Mt. Jiri
with nine observation stations. Three stations in bold are where the Parsivel and UVW measurements
were analyzed in this study. R1 and R2 show stations with a rain guage only.

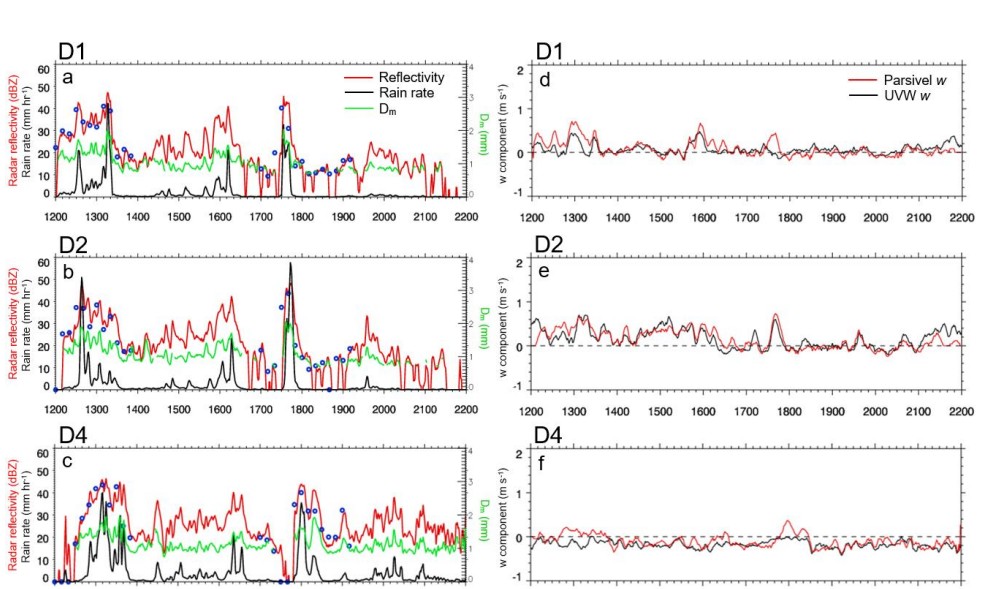

Figure 5. Time series of radar reflectivity (dBZ) in red line, rain rate (mm hr$^{-1}$) in black, and mass-weighted mean diameter (D$_m$, mm) in green at D1, D2, and D4 (left panels) and the time comparison of $w_{par}$ (m s$^{-1}$) in red and $w_{UVW}$ (m s$^{-1}$) in black at the same stations (right panels). Blue circles in the left panels indicate composite reflectivities (dBZ) from dual radars at the locations of D1, D2, and D4 for the selected periods.





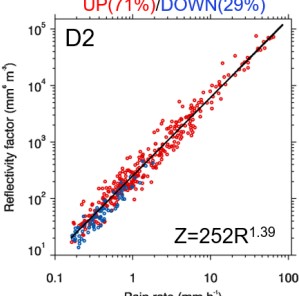
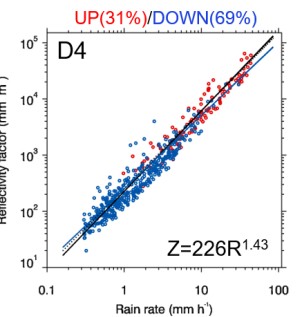

Figure 6. *Z-R* scatterplots at the three stations. Red dots indicate upward *w* and blue indicate downward *w*. Numbers on the top show percentages in each *w* category.




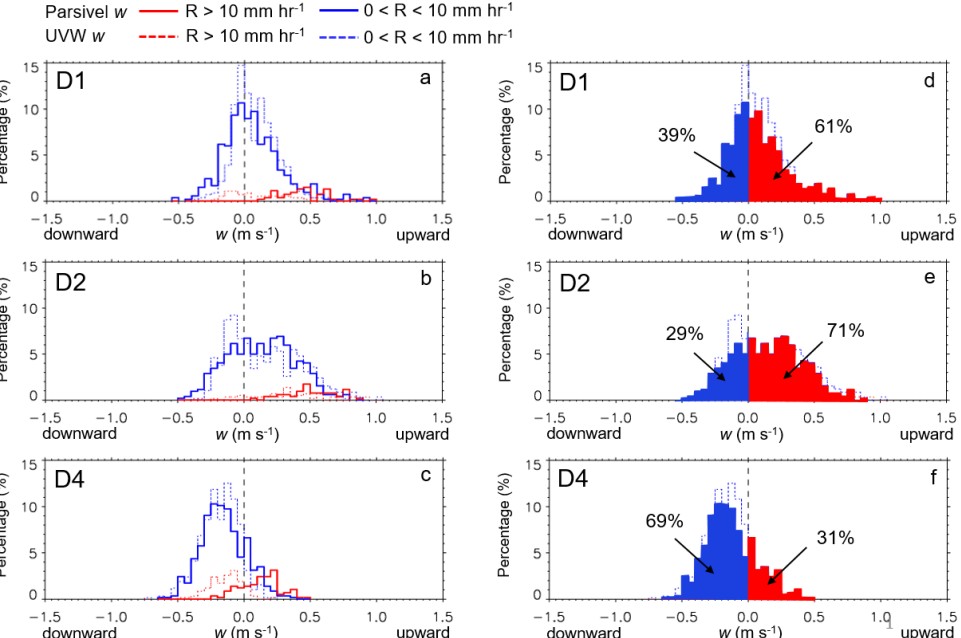

Figure 7. *w* histograms with regard to the two *R* groups (left) and those with percentages in the
upward and downward *w* groups at the three stations (right).






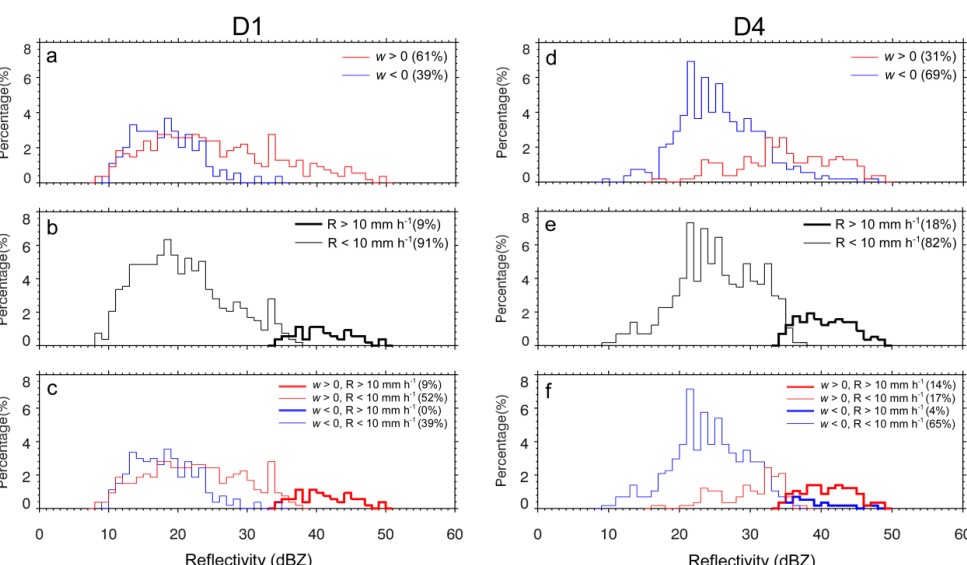

Figure 8. *Z* histograms with regard to *w*, *R*, and those in the four groups with percentage at D1 and D4.






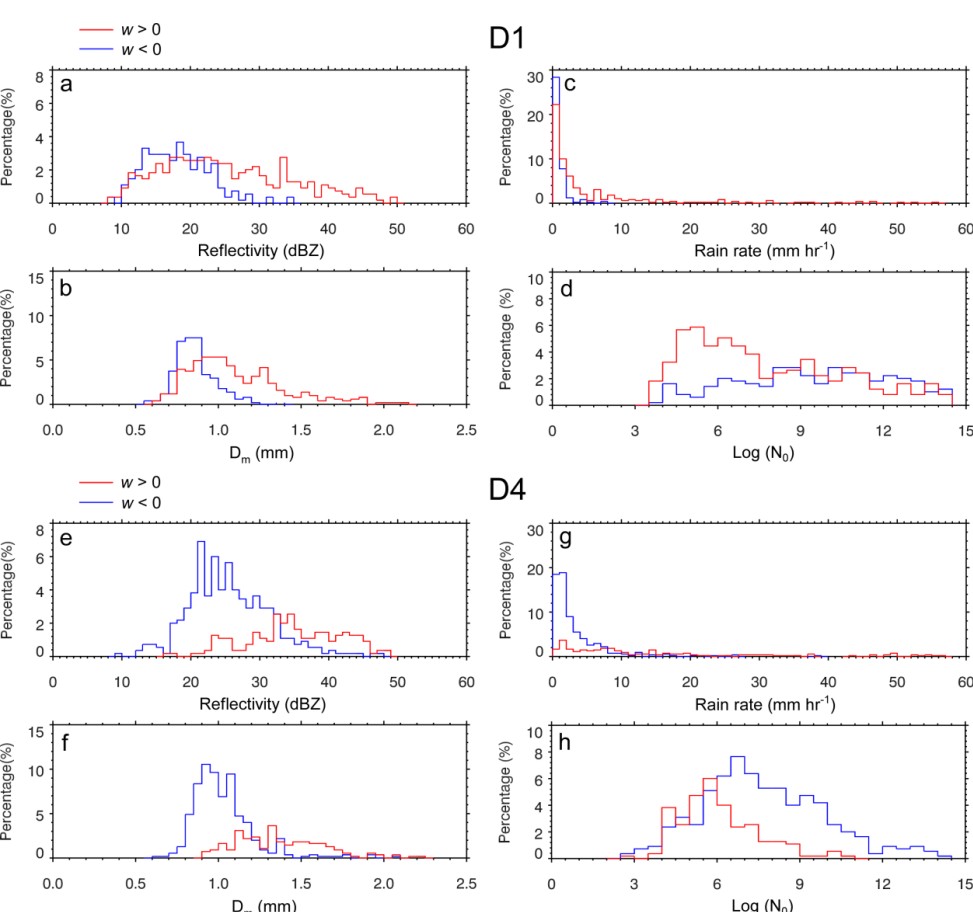

Figure 9. Histograms of retrieved DSD parameters with regard to the upward (red line) and downward
$w$ (blue): (a) radar reflectivity (dBZ), (b) rain rate (mm hr$^{-1}$), (c) $D_m$ (mm) and (d) $N_0$ in log scale at
D1 (top four panels) and the same as these but for D4 (bottom four panels).
