# Peer review of "Characteristics of vertical velocities estimated from drop size and fall velocity spectra of a Parsivel disdrometer"

_Atmospheric Measurement Techniques, 2018_

## Referee Comment (RC1) · Anonymous Referee #1 · 23 Mar 2018

This paper deals with the estimation of vertical (fall) velocities of rain drops from Parsivel measurements, assuming that the rain drop fall speeds follow the Gunn-Kinzer (G-K) variation (using Atlas et al. 1973 equation) and some minor altitude adjustment for air density. The estimates are compared with anemometer measurements in several different locations. Disdrometer data are also used to derive Z-R relationships and the mass-weighted mean diameter. The paper contains reasonable results and is suitable for publication in AMT, but some changes need to be made, as follows:

Line 162, and Fig. 2: It might be easier to follow this flow chart if some pertinent equations are added, corresponding to each step.

Lines 164-169, and the use of Fig. 3: Please note that although G-K formula is a good representation for the 'most probable' velocity - diameter variation (especially for > 1 mm drops), it is well known that for a given drop size, there will be a distribution of velocities associated with it (but narrow). For example see a very recent paper by Bringi et al., 2018: https://www.atmos-meas-tech.net/11/1377/2018/ in particular their Fig. 1 and 3.

Also note that for small drops, the most suitable formula is in Foote and DuToit (1969), eq. (10), with the coefficients given in their Table 1 for N=9.

Lines 186-189: The authors say "Also a dual-Doppler radar analysis (Liou et al., 2012) was also conducted to obtain 3-D wind components from radial velocity data of two Doppler radars as well as vertical structure of radar reflectivity in this mountain area." However, I don't see those results in the manuscript. It would be advisable to incorporate or make use of, at least qualitatively, velocity information from the dual-Doppler analyses.

Lines 217-218: "they just pointed up or downward, depending on w signs and magnitudes". This statement is not clear. Certainly, drop horizontal velocities will cause errors and some discussion on this needs to be included.

Please also note Appendix A of Thurai JAOT, 2017, along with their Fig. A1, which shows drop horizontal velocities, both in terms of magnitude and direction, derived from 2D video disdrometer measurements, and the excellent comparisons with the 10 m wind sensor data.

Last para in Section 4.1: Can the authors include a discussion on the role of DSDs in the calculated Z-R relationships?

Around lines 243-245: For R>10 mm/h, Fig. 7a shows very different histograms between Parsivel-based and wind-sensor based. This should be pointed out, and explained, if possible.

[Figure]

Lines 255-256: "In the downward w group, the largest percentage (69%) is found at D4 (Fig. 7f)" .. So why is this? Can this be explained?

Modest amount of revision is required.

Please also note the supplement to this comment:
https://www.atmos-meas-tech-discuss.net/amt-2018-63/amt-2018-63-RC1-supplement.pdf

---

## Referee Comment (RC2) · Anonymous Referee #2 · 10 Apr 2018

The paper "Characteristics of vertical velocities estimated from drop size and fall velocity spectra of a Parsivel disdrometer" by Kim and Song presents the results of an experimental study aimed to develop a technique to estimate the vertical velocity of raindrops in natural rain, and to apply the technique to convective precipitation around mount Jiri, in South Korea. The data from three measuring stations, equipped with a Parsivel disdrometer and an ultrasonic anemometer, are used to study the relation between the velocities measured by the two instruments in leeward and windward side of the mt. Jiri for convective and stratiform rain.

The paper is interesting, fairly well written and the topic is comprised among the subject

areas of AMT. I therefore suggest the publication of the paper, after a few modifications I suggest below.

Lines 154-160. The correction proposed by Authors to take into account the reduction of air density with altitude is around 1% for D4, and this error is largely negligible if compared to other experimental errors, so I suggest to cancel this discussion and to use the Atlas et al. (1973) relation. For example, ultrasonic anemometers show that the air velocity varies greatly (much more than 1%) at sub-minute scale, while the Authors assume the speed of air is constant (Vf) during one minute and different drops fall with different instantaneous velocity.

Figure 4. This is a 2-panel figure. On the left there is accumulated precipitation (color shades) but also isolines of altitude, I guess. Altitude is also reported on the right figure, enlarged, but the meaning of the color is not given. I suggest to simplify this figure, avoiding to repeat the same information twice, and better describing in the caption what is shown in the figure.

Figure 5 (a, b, c). The "composite reflectivity (dBZ) from the dual radar..." is never mentioned in the text and these data never used in the discussion: I suggest to remove the blue dots, and the sentence on lines 186-188.

Figure 5 (d, e, f). I suggest to expand the y-axis scale, say between -0.5 to 1 m s-1, in order to better appreciate the differences between the two vertical velocities.

Figure 5. Since it is discussed the coincidence of rainshowers and differences between the two w, it would probably better to put R/Z/Dm and w plots one above the other.

Lines 220-223. This sentence is not convincing and too speculative. The causes of increase or decrease of rainrate are very complex and cannot be understood by simply measure the point-like vertical velocity few tens of centimeters above the ground. What is measured here is not the updraft/downdraft of convective development (that cannot last for many hours), but probably the weak component of the wind speed due to the

uphill/downhill flux.

Lines 233-235. It is true that higher b indicates steeper relation between R and Z, bus does not tell anything about the "strength" of rainfall occurred, it is a measure of the relative occurrence of smaller and larger drops.

Line 242. It should be noted here that there are a plenty of algorithms based on DSD to discriminate convective and stratiform precipitation based on DSD and not only on rainrate (Tokay and Short, 1996, Caracciolo et al., 2006, Thomson et al., 2015, Thurai et al., 2016).

Figure 7. Please keep wpar and wUVW names as in the text and other figures. How are the histograms normalized? They are percent of what?

---

## Referee Comment (RC3) · Anonymous Referee #3 · 26 Apr 2018

Review of AMT-2018-63

By Dong-Kyun Kimand Chang-Keun Song

Manuscript title: Characteristics of vertical velocities estimated from drop size and fall velocity spectra of a Parsivel disdrometer.

This manuscript reports about the estimation of vertical air velocity by disdrometer (Parsivel) measurements. The estimation is based on the comparison between measured (by Parsivel) and theoretical vertical drop velocity. In particular, the mean measured drop velocity is calculated from Parsivel data. The estimated vertical air velocity is compared and validated with the vertical motion measured by a collocated ultrasonic anemometer. One case study, during the monsoon rainy season in South Korea, is analyzed at three different measurement sites. The characteristics of DSD parameters (i.e. radar reflectivity, rain rate, mean mass diameter, etc.) are analyzed with respect to the upward or downward estimated air motion.

The structure of the paper is linear, but at the same time many inaccuracies both scientific/descriptive and language (the paper should be checked by a native English) can be found within the paper. My main concern is related to the unsuitableness of the analyzed case study to validate the vertical air motion estimate from Parsivel measurements. The vertical air velocity mainly ranges between -0.5 and +0.5 ms$^{-1}$, but nothing is reported about the measurement uncertainty of the ultrasonic anemometer as well as the correction of the theoretical drop fall velocity due to the air density. Because of the very low values of vertical air motion, even during convective precipitation, the analysis carried out by the authors does not clarify the doubt that the vertical velocity estimates are within the measurement and process uncertainty. Due to these general considerations, I do not retain that the manuscript is ready to be published on the Atmospheric Measurement Techniques journal. I encourage the authors to deepen investigate the methodology by considering other case studies (involving higher values of vertical air velocity).

In the following, I report more specific comments.

- Line 108: the citations "Niu et al. (201)" and "Ulbrich (1992)" are not present in the reference list. Please, check all the reference list.

- Lines 116-118: referring to Tokay et al. (2009, 2014), how the Parsivel underestimation and overestimation of small and large drops, respectively, affects the calculation of the mean fall speed?

- Lines 142-146: what is the uncertainty of the ultrasonic anemometer measurements? This is a fundamental information needed to validate the air motion estimated from Parsivel data.

- Equation 2: is there a meteorological station (able to measure air pressure and temperature) collocated with the Parsivel and ultrasonic anemometer? This can be useful in a better quantification of the deviation from the drop fall velocity at sea level and in quantifying the difference between using the standard atmosphere equation and the measured temperature and pressure.

- Lines 159-160: there is no correspondence between what the authors say in the abstract (and within the text), that is the field observational site is on the Mt. Jiri at 1915 m above the sea level, and what reported here, that is the three measurement sites are at very lower altitudes. Please, uniform the information about the field observational sites.

- Lines 186-191: the authors cite about the analysis of 3D wind components as well as the vertical structure of the precipitation from dual-Doppler radar measurements, but data are not shown neither discussed. Please add an analysis on this or remove the statement.
They also refer to "..a daily accumulated accumulated rainfall distribution.." but they refer to the case study (as correctly reported in the caption of Figure 4). Please, uniform the text to avoid misunderstandings in the reader.

- Figure 5 and relative discussion: I agree with the authors that the trend and Parsivel w and UVW w is similar but, as already reported in the introductive part, the very low values along the whole period cannot be useful the validate the procedure, in my opinion.

- Why Figure 6 for D4 shows three different fit lines? Do they refer to UP, DOWN and UP/DOWN together data? If it is the case, this should be mentioned in the next. Do they overlap at D1 and D2 site?

- Lines 240-242: there are several more recent paper (Tokay and Short, 196, Bringi et al., 2003, niu et al., 2010 just to make a few examples) reporting different methodologies to discriminate stratiform and convective precipitation rather than a simple rain rate threshold.

- Lines 324-330: in my opinion this is a too strong speculation. The technique is surely promising but has to be tested in different conditions (i.e. more intense vertical winds) or the authors have to more discuss about the sensitivity of the ultrasonic anemometer used to validate the technique. "notably different characteristics in magnitude and signs and signs between the windward and leeward side…" are not so evident and in contradiction with what the authors report just below this sentence where they state the vertical wind range between -0.5 and +0.5 ms$^{-1}$ for the case study.

---

## Author Comment (AC1) · 10 Jun 2018

This paper deals with the estimation of vertical (fall) velocities of rain drops from Parsivel measurements, assuming that the rain drop fall speeds follow the Gunn-Kinzer (G-K) variation (using Atlas et al. 1973 equation) and some minor altitude adjustment for air density. The estimates are compared with anemometer measurements in several different locations. Disdrometer data are also used to derive Z-R relationships and the mass-weighted mean diameter. The paper contains reasonable results and is suitable for publication in AMT, but some changes need to be made, as follows:

Line 162, and Fig. 2: It might be easier to follow this flow chart if some pertinent equations are added, corresponding to each step.

➔ Corresponding equations at each step were added to the figure 2.

Lines 164-169, and the use of Fig. 3: Please note that although G-K formula is a good representation for the 'most probable' velocity - diameter variation (especially for > 1 mm drops), it is well known that for a given drop size, there will be a distribution of velocities associated with it (but narrow). For example see a very recent paper by Bringi et al., 2018: https://www.atmos-meas-tech.net/11/1377/2018/ in particular their Fig. 1 and 3.

➔ Thanks for the paper. We agree that there is a distribution of velocities for a given drop size. Also, $V_p$ and $V_f$ in Eq (2) that were used to calculate w are mean values for a given one minute D-$V_p$ spectrum. Please see the equations about how to get mean $V_p$ and $V_f$ in Fig. 2. So their variations over a given range of drop size are minimized.

Also note that for small drops, the most suitable formula is in Foote and DuToit (1969), eq. (10), with the coefficients given in their Table 1 for N=9.

➔ Thanks for the paper and formula. We will consider to test the equation (10) with the coefficients in the future works.

Lines 186-189: The authors say "Also a dual-Doppler radar analysis (Liou et al., 2012) was also conducted to obtain 3-D wind components from radial velocity data of two Doppler radars as well as vertical structure of radar reflectivity in this mountain area."

However, I don't see those results in the manuscript. It would be advisable to incorporate or make use of, at least qualitatively, velocity information from the dual-Doppler analyses.

➔ Actually the composite reflectivity values from dual-Doppler radars were plotted in Figures 5a,c,e. They are reflectivities, not retrieved vertical velocities. We deals with very surface measurements of velocities (Parsivel and anemometer). We have not analyzed vertical velocities retrieved from dual-Doppler analyses (it starts above several hundred meters). We have examined this as well but found that there were large differences since there were large errors and uncertainties in vertical velocities retrieved from dual-Doppler radars. One other reviewer also mentioned about this issue, saying that they don't see their analysis in the manuscript.

So we removed the sentences regarding the dual-Doppler radar analysis (also the radar composite reflectivities in blue dots were removed in Figure 5a,c,e.)

Lines 217-218: "they just pointed up or downward, depending on w signs and magnitudes". This statement is not clear. Certainly, drop horizontal velocities will cause errors and some discussion on this needs to be included.

➔ We meant that the magnitudes of horizontal winds were much larger than those of vertical velocities. Thus, they are almost horizontal and just head upward or downward slightly with signs (plus/minus). We changed the sentences as follows.

"winds are almost horizontal during the whole period and they point upward or downward slightly with the *w* signs."

Also as shown in Fig.5, both the estimated w (w$_{par}$) and measured w (w$_{UVW}$) are very low in magnitude. As you know, these are just a vertical component of winds. Therefore, on the other hand, the low w values and stronger horizontal winds almost 5 times larger than the measured w (not shown in this manuscript) indicate that the winds just head up and down slightly with w signs. For larger rainfall (larger Z), retrieved w values were found higher, meaning that there were slightly upward-pointing large scale flow (even near the surface) around the mountain, probably producing converging-upward air and strengthening the orographic rain system. So we found that even very slightly upward motions can make favorable conditions for increasing Z and R in these mountain areas. Again, we need to test the disdrometer-based technique in other places and events. Also these w results are obtained at surface, not aloft. For the vertical extent of up/downdrafts, there is a need to examine further by using small vertically pointing radar (like micro rain radar) or profiler observations in the future.

Plus, UVW measures airflow itself but Parsivel measures particle movements along the airflow in the sampling area. Drops in different mass (small/large) responses to the same airflow differently. These are very complex and difficult for us to discriminate even if we have Parsivel and UVW observation data. We are preparing for another manuscript in relation to factors like winds (wind shear) soon.

We included these words and explanation in the summary and conclusion section of the revised version (page 14) as follows.

"Eventually the newly developed technique that estimates w values from Parsivel drop size and fall velocity spectra is found physically meaningful although it needs to be further tested in other places and events. It would be applicable to w retrieval and comparison studies near the surface to investigate rain microphysics associated with up-/downward motions. The different w percentages at the different locations stressed their dependence on observed D-Vp distributions which vary largely as a result of complex factors such as rainfall intensity, up-/downdrafts, wind speed, turbulence, and so on.

In this study, both the observed and estimated w values were very small in magnitude mostly between -0.5 and +0.5 m s$^{-1}$, about one fifth of the measured horizontal wind speeds. As known, the w values are just a vertical component of winds. Thus the low w values indicate almost horizontal winds that just head up and down slightly with the w signs. During the high *R* periods, the estimated w values were larger in a positive sign (windward side), suggesting that there were slightly upward flows around the mountain. Probably this produces an environment of converging-upward air in large scale and helps to intensify the orographic rain system, increasing Z and R."

Please also note Appendix A of Thurai JAOT, 2017, along with their Fig. A1, which shows drop horizontal velocities, both in terms of magnitude and direction, derived from 2D video disdrometer measurements, and the excellent comparisons with the 10m wind sensor data.

Last para in Section 4.1: Can the authors include a discussion on the role of DSDs in the calculated Z-R relationships?

➔ For given Z, there were relatively stronger rainfall in the leeward side (D4). We modified the sentences as follows

"Power-law $Z$-$R$ relations at a form of $Z=\alpha R^{\beta}$ are compared between the observation sites in Fig. 6. There was a decrease in the coefficient $\alpha$ from D1 and D2 (250, 252) on the windward side to D4 (226) on the leeward side. The exponent $\beta$ did not show notable change between the sides. The noticeable decrease in $\alpha$ suggests that for a given $Z$, $R$ is larger at D4 than D1 and D2. This is consistent to histograms of DSD parameters in the later section showing the larger mean $R$ and $D_m$ at D4."

Around lines 243-245: For R>10 mm/h, Fig. 7a shows very different histograms between Parsivel-based and wind-sensor based. This should be pointed out, and explained, if possible.

➜ According to our analysis (for other cases) with regard to winds and reflectivities measured by Parsivel, strong winds tend to make a downward spread of drop fall velocities in Parsivel-measured drop and fall velocity spectra. Mathematically, this downward spread decreases Parsivel-measured drop fall velocities (decrease in $V_p$ in Eqn(2) in the text) and hence $w_{par}$ becomes more positive, making larger difference with $w_{UVW}$. As you know, environmental winds are very important for accurate studies of retrieving $w$ from Parsivel since relatively small (tiny) drops can be blown more easily along with winds but this was not fully considered in this study as we mentioned in the manuscript. In association with winds and rain intensities, we are preparing for another manuscript for other rain cases in these areas and will submit it soon. We added the following sentences.

"The relatively larger difference between the $w_{par}$ and $w_{UVW}$ histograms is found in the convective class of D1 and this is likely due to strong wind speeds that tend to make a downward spread in measured $D$ vs. $V_p$ spectra of Parsivel. Mathematically, this downward spread decreases Parsivel-measured drop fall velocities (i.e., decrease in $V_p$ in Eq (2)) and hence $w_{par}$ becomes more positive, making a larger difference with $w_{UVW}$."

Lines 255-256: "In the downward w group, the largest percentage (69%) is found at D4 (Fig. 7f)" .. So why is this? Can this be explained?

➜ The Parsivel measurements were made in mountain areas. D4 is located in the leeward side and the other two sites (D1, D2) are located in the windward side of the mountain. As the system moves from the south to north of the mountain, upward motions prevailed in the windward while downward motions were more dominant in the leeward side (but there were upward motions as well). By the percentage of downward w groups at the three sites, it was largest at D4. We removed the sentence and added the followings.

"The colored areas with the percentages show readily which $w$ group is far dominant. As noted, upward motions were dominant at D1 and D2 while downward motions were dominant at D4. However, they did not show large percentage differences at all the sites, suggesting that either upward or downward motions have not happened overwhelmingly in this event."

---

## Author Comment (AC2) · 10 Jun 2018

The paper "Characteristics of vertical velocities estimated from drop size and fall velocity spectra of a Parsivel disdrometer" by Kim and Song presents the results of an experimental study aimed to develop a technique to estimate the vertical velocity of raindrops in natural rain, and to apply the technique to convective precipitation around mount Jiri, in South Korea. The data from three measuring sites, equipped with a Parsival disdrometer and an ultrasonic anemometer, are used to study the relation between the velocities measured by the two instruments in leeward and windward side of the mt. Jiri for convective and stratiform rain.

The paper is interesting, fairly well written and the topic is comprised among the subject areas of AMT. I therefore suggest the publication of the paper, after a few modifications
I suggest below.

Lines 154-160. The correction proposed by Authors to take into account the reduction of air density with altitude is around 1% for D4, and this error is largely negligible if compared to other experimental errors, so I suggest to cancel this discussion and to use the Atlas et al. (1973) relation. For example, ultrasonic anemometers show that the air velocity varies greatly (much more than 1%) at sub-minute scale, while the Authors assume the speed of air is constant (Vf) during one minute and different drops fall with different instantaneous velocity.

➔ Yes, we removed the correction term in Eq (2), just using the original relation of Atlas et al.(1973) and modified the sentences as below. Also, $V_p$ and $V_f$ in Eq (2) that were used to calculate w are mean values for a given one minute D-$V_p$ spectrum (1 min interval). Please see the equations about how to get mean $V_p$ and $V_f$ in Fig. 2.

"Altitudes of D1, D2, and D4 are 105, 280 and 313 m ASL, respectively. Due to the very low altitudes of these observation sites, change in atmospheric density with height is negligible and thus the atmospheric density correction (Beard, 1985) on $V_f$ is ignored."

Figure 4. This is a 2-panel figure. On the left there is accumulated precipitation (color shades) but also isolines of altitude, I guess. Altitude is also reported on the right figure, enlarged, but the meaning of the color is not given. I suggest to simplify this figure, avoiding to repeat the same information twice, and better describing in the caption what is shown in the figure.

➔ Yes we understand what you pointed out here. To be exact, the contours are not the same in both the panels. On the left panel, they are contours of altitude at 300 m interval with accumulation rainfall amounts in color, just giving broad information on where more rainfall has occurred around the mountain (i.e., there was relatively more rainfall south of the mountain) in a much larger domain. On the right panel, contours of altitude are plotted at 200 m interval. This time, we added a color bar to the next that shows altitudes and modified the figure caption as follows.

"Figure 4. (a) Distribution of an acumulated rainfall (mm) on 1 July over contours of altitude at 300 m interval and (b) the enlarged topography of Mt. Jiri with contours of altitude at 200 m interval, showing nine observation sites. Three sites in red are where the Parsivel and UVW measurements were analyzed in this study. R1 and R2 show sites with a rain gauge only"

Figure 5 (a, b, c). The "composite reflectivity (dBZ) from the dual radar: : :" is never mentioned in the text and these data never used in the discussion: I suggest to remove the blue dots, and the sentence on lines 186-188.

➔ The blue dots and related sentences were removed.

Figure 5 (d, e, f). I suggest to expand the y-axis scale, say between -0.5 to 1 m s-1, in order to better appreciate the differences between the two vertical velocities.

➔ The plots were modified as you suggested.

Figure 5. Since it is discussed the coincidence of rainshowers and differences between the two w, it would probably better to put R/Z/Dm and w plots one above the other.

➔ The plots were modified as you suggested. (We were concerned about the figure setting a little bit, though.)

Lines 220-223. This sentence is not convincing and too speculative. The causes of increase or decrease of rainrate are very complex and cannot be understood by simply measure the point-like vertical velocity few tens of centimeters above the ground. What is measured here is not the updraft/downdraft of convective development (that cannot last for many hours), but probably the weak component of the wind speed due to the uphill/downhill flux.

➔ Here, our analysis depends entirely on surface measurements of Parsivel and UVW and estimated w values from Parsivel data. With these surface measurements, it may be difficult to relate them to up/downdrafts aloft. However, certainly, there was an increase of R within downward motions (negative $w_{UVW}$) around 1300 and 1630 LST in Fig. 5. Due to the very low w magnitude, I do not call this pronounced downdraft (maybe related to downdrafts aloft but we can't tell with surface measurements only). As I wrote below, I think they are slightly downward-pointing airflow in large scale (induced by the rainfall system and mountain) (please read the paragraph below). Also if you look at Figs. 8 and 9 (with Fig. 5), the histograms of parameters make sense with our knowledge in regard to convective and stratiform rainfall. In this study, they were obtained with additional w information, which is very meaningful and promising. As you pointed out, reasons for the R increase and decrease may be complex in mountainous areas. There is a need to test the disdrometer-based technique in other places and events to generalize.

As shown in Fig.5, both the estimated w ($w_{par}$) and measured w ($w_{UVW}$) are very low in magnitude. As you know, these are just a vertical component of winds. Therefore, on the other hand, the low w values and stronger horizontal winds almost 5 times larger than the measured w (not shown in this manuscript) indicate that the winds just head up and down slightly with w signs. For larger rainfall (larger Z), retrieved w values were found higher, meaning that there were slightly upward-pointing large scale flow (even near the surface) around the mountain, probably producing converging-upward air and strengthening the orographic rain system. So we found that even very slightly upward motions can make favorable conditions for increasing Z and R in these mountain areas. Again, we need to test the disdrometer-based technique in other places and events. Also these w results are obtained at surface, not aloft. For the vertical extent of up/downdrafts, there is a need to examine further by using small vertically pointing radar (like micro rain radar) or profiler observations in the future.

Plus, UVW measures airflow itself but Parsivel measures particle movements along the airflow in the sampling area. Drops in different mass (small/large) responses to the same airflow differently. These are very complex and difficult for us to discriminate even if we have Parsivel and UVW observation data. We are preparing for another manuscript in relation to factors like winds (wind shear) soon.

We included these words and explanation in the summary and conclusion section of the revised version (page 14) as follows.

"Eventually the newly developed technique that estimates w values from Parsivel drop size and fall velocity spectra is found physically meaningful although it needs to be further tested in other places and events. It would be applicable to w retrieval and comparison studies near the surface to investigate rain microphysics associated with up-/downward motions. The different w percentages at the different locations stressed their dependence on observed D-Vp distributions which vary largely as a result of complex factors such as rainfall intensity, up-/downdrafts, wind speed, turbulence, and so on.

In this study, both the observed and estimated w values were very small in magnitude mostly between -0.5 and +0.5 m s$^{-1}$, about one fifth of the measured horizontal wind speeds. As known, the w values are just a vertical component of winds. Thus the low w values indicate almost horizontal winds that just head up and down slightly with the w signs. During the high $R$ periods, the estimated w values were larger in a positive sign (windward side), suggesting that there were slightly upward flows around the mountain. Probably this produces an environment of converging-upward air in large scale and helps to intensify the orographic rain system, increasing Z and R."

Lines 233-235. It is true that higher b indicates steeper relation between R and Z, bus does not tell anything about the "strength" of rainfall occurred, it is a measure of the relative occurrence of smaller and larger drops.

➔ Yes I agree. This is about the comparison of Z-R relations between two different places. For a given reference value of Z, we can tell larger R or small R. Often times, Z-R comparisons are used to see a relative strength between convective and stratiform rain for a given Z (please see Yuter and Houze (1997) and Atlas et al. (2000)). We modified the sentences as follows.

"Power-law $Z$-$R$ relations at a form of $Z=\alpha R^{\beta}$ are compared between the observation sites in Fig. 6. There was a decrease in the coefficient $\alpha$ from D1 and D2 (250, 252) on the windward side to D4 (226) on the leeward side. The exponent $\beta$ did not show notable change between the sides. The noticeable decrease in $\alpha$ suggests that for a given $Z$, $R$ is larger at D4 than D1 and D2. This is consistent to histograms of DSD parameters in the later section showing the larger mean $R$ and $D_m$ at D4."

Line 242. It should be noted here that there are a plenty of algorithms based on DSD to discriminate convective and stratiform precipitation based on DSD and not only on rainrate (Tokay and Short, 1996, Caracciolo et al., 2006, Thomson et al., 2015, Thurai et al., 2016).

➔ Yes we added more references as follows.

"In this study, a simple $R$ threshold, $R < 10$ mm h$^{-1}$ and $R > 10$ mm h$^{-1}$ (Leary and Houze 1979; Testud et al., 2001), to discriminate stratiform and convective rain was used although there have been a plenty of other methods based on DSDs and vertical profiles to discriminate stratiform and convective rain (Bringi et al., 2003; Caracciolo et al., 2006; Thompson et al., 2015; Thurai et al., 2016; Tokay and Short 1996; Tokay et al., 1999; Ulbrich and Atlas 2002; Williams et al., 1995)."

Figure 7. Please keep wpar and wUVW names as in the text and other figures. How are the histograms normalized? They are percent of what?

➔ We keep them to be consistent in the text and figures. They were not normalized. They are percent of frequency of occurrences. These are histograms with a bin size of 0.05 m s$^{-1}$.

Occurrences in each w group were changed to percent values as they are divided by the total occurrence during the analysis period. So if we add them up, it amounts to 100%. So they are percent values of frequency of occurrence. We added sentences to be clarified as follows.

"Occurrences of upward and downward motions were changed to percentage values as they are divided by a total count of upward and downward $w$ during the entire period. A bin size for these histograms is 0.05 m s$^{-1}$."

---

## Author Comment (AC3) · 10 Jun 2018

Review of AMT-2018-63
By Dong-Kyun Kim and Chang-Keun Song
Manuscript title: Characteristics of vertical velocities estimated from drop size and fall velocity
spectra of a Parsivel disdrometer.

   This manuscript reports about the estimation of vertical air velocity by disdrometer (Parsivel)
measurements. The estimation is based on the comparison between measured (by Parsivel) and
theoretical vertical drop velocity. In particular, the mean measured drop velocity is calculated from
Parsivel data. The estimated vertical air velocity is compared and validated with the vertical motion
measured by a collocated ultrasonic anemometer. One case study, during the monsoon rainy season in
South Korea, is analyzed at three different measurement sites. The characteristics of DSD parameters
(i.e. radar reflectivity, rain rate, mean mass diameter, etc.) are analyzed with respect to the upward or
downward estimated air motion.
   The structure of the paper is linear, but at the same time many inaccuracies both
scientific/descriptive and language (the paper should be checked by a native English) can be found
within the paper. My main concern is related to the unsuitableness of the analyzed case study to
validate the vertical air motion estimate from Parsivel measurements. The vertical air velocity mainly
ranges between -0.5 and +0.5 ms-1, but nothing is reported about the measurement uncertainty of the
ultrasonic anemometer as well as the correction of the theoretical drop fall velocity due to the air
density. Because of the very low values of vertical air motion, even during convective precipitation,
the analysis carried out by the authors does not clarify the doubt that the vertical velocity estimates are
within the measurement and process uncertainty. Due to these general considerations, I do not retain
that the manuscript is ready to be published on the Atmospheric Measurement Techniques journal. I
encourage the authors to deepen investigate the methodology by considering other case studies
(involving higher values of vertical air velocity). In the following, I report more specific comments.

- Line 108: the citations "Niu et al. (201)" and "Ulbrich (1992)" are not present in the reference list.
Please, check all the reference list.

   ➔   All the references were carefully checked.

- Lines 116-118: referring to Tokay et al. (2009, 2014), how the Parsivel underestimation and
overestimation of small and large drops, respectively, affects the calculation of the mean fall speed?

   ➔   Our Parsivels are the old version ones. We did not investigate the underestimation and
       overestimation of small and large drops quantitatively and their effects on the mean fall
       velocities. In our measurements, we found that the mean drop fall velocities measured by
       Parsivel are much more sensitive to fall velocities of smaller drops ranged from 0.25 mm
       (almost the minimum size detected by Parsivel) to about 2.0 mm than those of large drops (>
       2.0 mm). More works need to be done for this issue.

- Lines 142-146: what is the uncertainty of the ultrasonic anemometer measurements? This is a
fundamental information needed to validate the air motion estimated from Parsivel data.

   ➔   The accuracies of the instrument for wind speed and direction were added as follows.

   "The accuracies are $\pm 0.05$ m s$^{-1}$ for wind speed (0 to 30 m s$^{-1}$) and $\pm 2$ degrees for wind direction
   (0 to 30 m s$^{-1}$), respectively."

- Equation 2: is there a meteorological site (able to measure air pressure and temperature) collocated
with the Parsivel and ultrasonic anemometer? This can be useful in a better quantification of the

deviation from the drop fall velocity at sea level and in quantifying the difference between using the standard atmosphere equation and the measured temperature and pressure.

➔ There are meteorological sites around the mountains but no sensors to measure pressure and temperature in the collocated instrument sites. One other reviewer suggested to remove the atmospheric density correction term (Beard, 1985) since the site altitudes are very low and related errors in drop fall velocity are negligible.

So we removed that part in the equation and modified the text as follows.

"Altitudes of D1, D2, and D4 are 105, 280 and 313 m ASL, respectively. Due to the very low altitudes of these observation sites, change in atmospheric density with height is negligible and thus the atmospheric density correction (Beard, 1985) on $V_f$ is ignored."

- Lines 159-160: there is no correspondence between what the authors say in the abstract (and within the text), that is the field observational site is on the Mt. Jiri at 1915 m above the sea level, and what reported here, that is the three measurement sites are at very lower altitudes. Please, uniform the information about the field observational sites.

➔ All the site altitudes are above sea level. AGL was changed to ASL.

- Lines 186-191: the authors cite about the analysis of 3D wind components as well as the vertical structure of the precipitation from dual-Doppler radar measurements, but data are not shown neither discussed. Please add an analysis on this or remove the statement.

➔ We removed the sentences as other reviewers also commented similarly on this.

They also refer to "..a daily accumulated accumulated rainfall distribution.." but they refer to the case study (as correctly reported in the caption of Figure 4). Please, uniform the text to avoid misunderstandings in the reader.

➔ We changed the text to avoid confusion as follows.

"Figure 4 shows a distribution of accumulated rainfall on 1 July"

- Figure 5 and relative discussion: I agree with the authors that the trend and Parsivel w and UVW w is similar but, as already reported in the introductive part, the very low values along the whole period cannot be useful the validate the procedure, in my opinion.

➔ As shown in Fig.5, both the estimated w (w_par) and measured w (w_UVW) are very low in magnitude. As you know, these are just a vertical component of winds. Therefore, on the other hand, the low w values and stronger horizontal winds almost 5 times larger than the measured w (not shown in this manuscript) indicate that the winds just head up and down slightly with w signs. For larger rainfall (larger Z), retrieved w values were found higher, meaning that there were slightly upward-pointing large scale flow (even near the surface) around the mountain, probably producing converging-upward air and strengthening the orographic rain system. So we found that even very slightly upward motions can make favorable conditions for increasing Z and R in these mountain areas. Again, we need to test the disdrometer-based technique in other places and events. Also these w results are obtained at surface, not aloft. For the vertical extent of up/downdrafts, there is a need to examine further by using small vertically pointing radar (like micro rain radar) or profiler observations in the future.

Plus, UVW measures airflow itself but Parsivel measures particle movements along the airflow in the sampling area. Drops in different mass (small/large) responses to the same

airflow differently. These are very complex and difficult for us to discriminate even if we have Parsivel and UVW observation data. We are preparing for another manuscript in relation to factors like winds (wind shear) soon.

We included these words and explanation in the summary and conclusion section of the revised version (page 14) as follows.

➔     "Eventually the newly developed technique that estimates w values from Parsivel drop size and fall velocity spectra is found physically meaningful although it needs to be further tested in other places and events. It would be applicable to w retrieval and comparison studies near the surface to investigate rain microphysics associated with up-/downward motions. The different w percentages at the different locations stressed their dependence on observed D-Vp distributions which vary largely as a result of complex factors such as rainfall intensity, up-/downdrafts, wind speed, turbulence, and so on.

   In this study, both the observed and estimated w values were very small in magnitude mostly between -0.5 and 0.5 m s$^{-1}$, about one fifth of the measured horizontal wind speeds. As known, the w values are just a vertical component of winds. Thus the low w values indicate almost horizontal winds that just head up and down slightly with the w signs. During the high $R$ periods, the estimated w values were larger in a positive sign (windward side), suggesting that there were slightly upward flows around the mountain. Probably this produces an environment of converging-upward air in large scale and helps to intensify the orographic rain system, increasing Z and R."

- Why Figure 6 for D4 shows three different fit lines? Do they refer to UP, DOWN and UP/DOWN together data? If it is the case, this should be mentioned in the next. Do they overlap at D1 and D2 site?

➔   Thanks for pointing this out. Yes the three fit lines are given for up, down, and up/down all data. For consistency with Figs 6a and 6b, we modified Figure 6c only with one fit line for all w data. (other two lines were removed) Each coefficient $\alpha$ and exponent $\beta$ in the text were obtained from these lines.

- Lines 240-242: there are several more recent paper (Tokay and Short, 196, Bringi et al., 2003, niu et al., 2010 just to make a few examples) reporting different methodologies to discriminate stratiform and convective precipitation rather than a simple rain rate threshold.

➔   Yes, there have been many methods regarding discriminating stratiform and convective rainfall. The simple R threshold used in this study is just the simplest one of them. But which method that classifies rain type is used is not really important here because similar w patterns (showing the w similarity) would be expected from any method although percentages in each w group will be changed a little bit, though.

We added more references as follows.

"In this study, a simple $R$ threshold, $R < 10$ mm h$^{-1}$ and $R > 10$ mm h$^{-1}$ (Leary and Houze 1979; Testud et al., 2001), to discriminate stratiform and convective rain was used although there have been a plenty of other methods based on DSDs and vertical profiles to discriminate stratiform and convective rain (Bringi et al., 2003; Caracciolo et al., 2006; Thompson et al., 2015; Thurai et al., 2016; Tokay and Short 1996; Tokay et al., 1999; Ulbrich and Atlas 2002; Williams et al., 1995)."

- Lines 324-330: in my opinion this is a too strong speculation. The technique is surely promising but has to be tested in different conditions (i.e. more intense vertical winds) or the authors have to more discuss about the sensitivity of the ultrasonic anemometer used to validate the technique.

"notably different characteristics in magnitude and signs and signs between the windward and leeward side…" are not so evident and in contradiction with what the authors report just below this sentence where they state the vertical wind range between -0.5 and +0.5 ms-1 for the case study.

➔ Yes, we agree with your point and we need to test the technique in different places and conditions. We modified the sentences as follows.

"Eventually the newly developed technique that estimates w values from Parsivel drop size and fall velocity spectra is found physically meaningful although it needs to be further tested in other places and events. It would be applicable to w retrieval and comparison studies near the surface to investigate rain microphysics associated with up-/downward motions. The different w percentages at the different locations stressed their dependence on observed D-Vp distributions which vary largely as a result of complex factors such as rainfall intensity, up-/downdrafts, wind speed, turbulence, and so on.

In this study, both the observed and estimated w values were very small in magnitude mostly between -0.5 and 0.5 m s$^{-1}$, about one fifth of than the measured horizontal wind speeds. As known, the w values are just a vertical component of winds. Thus the low w values indicate almost horizontal winds that just head up and down slightly with the w signs. During the high $R$ periods, the estimated w values were larger in a positive sign (windward side), suggesting that there were slightly upward flows around the mountain. Probably this produces an environment of converging-upward air in large scale and helps to intensify the orographic rain system, increasing Z and R."

---

## Author Response (AR1)

| 1           |                                                                                                                                                                                                                                                                                                                                                                                                                                                                                                                                                                                                                                                                                                                                                                                                                                                                                                                                                                                                  |
|-------------|--------------------------------------------------------------------------------------------------------------------------------------------------------------------------------------------------------------------------------------------------------------------------------------------------------------------------------------------------------------------------------------------------------------------------------------------------------------------------------------------------------------------------------------------------------------------------------------------------------------------------------------------------------------------------------------------------------------------------------------------------------------------------------------------------------------------------------------------------------------------------------------------------------------------------------------------------------------------------------------------------|
| 2           |                                                                                                                                                                                                                                                                                                                                                                                                                                                                                                                                                                                                                                                                                                                                                                                                                                                                                                                                                                                                  |
| 3           |                                                                                                                                                                                                                                                                                                                                                                                                                                                                                                                                                                                                                                                                                                                                                                                                                                                                                                                                                                                                  |

[revised manuscript text omitted]